# A Review of T-Cell Related Therapy for Osteosarcoma

**DOI:** 10.3390/ijms21144877

**Published:** 2020-07-10

**Authors:** Kazushige Yoshida, Masanori Okamoto, Kaoru Aoki, Jun Takahashi, Naoto Saito

**Affiliations:** 1Columbia Stem Cell Initiative, Columba University, 650 West 168th Street, BB1108, New York, NY 10032, USA; ky2432@cumc.columbia.edu; 2Department of Orthopaedic Surgery, Shinshu University School of Medicine, 3-1-1 Asahi, Matsumoto, Nagano 390-8621, Japan; ryouyuma@shinshu-u.ac.jp (M.O.); jtaka@shinshu-u.ac.jp (J.T.); 3Physical Therapy Division, School of Health Sciences, Shinshu University,3-1-1 Asahi, Matsumoto, Nagano 390-8621, Japan; kin29men@shinshu-u.ac.jp; 4Institute for Biomedical Sciences, Interdisciplinary Cluster for Cutting Edge Research, Shinshu University, 3-1-1 Asahi, Matsumoto, Nagano 390-8621, Japan

**Keywords:** osteosarcoma, immunotherapy, T-cell

## Abstract

Osteosarcoma is one of the most common primary malignant tumors of bone. The combination of chemotherapy and surgery makes the prognosis better than before, but therapy has not dramatically improved over the last three decades. This is partially because of the lack of a novel specialized drug for osteosarcoma, which is known as a tumor with heterogeneity. On the other hand, immunotherapy has been one of the most widely used strategies for many cancers over the last ten years. The therapies related to T-cell response, such as immune checkpoint inhibitor and chimeric antigen receptor T-cell therapy, are well-known options for some cancers. In this review, we offer the accumulated knowledge of T-cell-related immunotherapy for osteosarcoma, and discuss the future of the therapy.

## 1. Introduction

Osteosarcoma is one of the most common primary malignant tumors of bone. The tumor occurs predominantly in adolescents, with a second peak amongst older adults [1]. The standard therapy for osteosarcoma is surgery to excise the tumor with an appropriate margin combined with pre- and post-operative chemotherapy [2]. This combined therapy improves the 5-year survival rate to 60–78% in patients with localized disease [3], but it means the presence of non-curative patients and it seems to have not improved over the past three decades. One reason for this is that the drugs used for the chemotherapy mainly consist of traditional ones such as cisplatin, doxorubicin, ifosfamide, and methotrexate [4]. There were some attempts to expand the indication of drugs for osteosarcoma therapy [5,6], but attempts to create new drugs, such as osteosarcoma specific molecular targeted drugs, have not necessarily been successful [7]. The heterogeneity of osteosarcoma [8,9] is thought to be one of the reasons for this difficulty.

On the other hand, immunotherapy has been one of the most focused on strategies for many cancers over the last ten years. The therapies related to T-cell response, like immune checkpoint inhibitor (ICI) [10] or chimeric antigen receptor (CAR) T-cell therapy [11], are already known as good options for some cancers. For osteosarcoma especially, these therapeutic options are promising as it has been reported that the number of tumor infiltrating T-cells is greater than that of other types of sarcoma [12]. Because of this, many immune therapies are being trialed in pre- and post-clinical settings.

In this review, we offer the accumulated knowledge of T-cell related immunotherapy for osteosarcoma and discuss its future.

## 2. Cancer Immune Therapy and Cancer Immunoediting

The immune system distinguishes between the self and non-self and eliminates the non-self. There are many factors involved in maintaining the immune system. Immunotherapy broadly means therapy using this system or its components. The first trial of immunotherapy for cancer was organized by Coley, known as an expert surgeon for malignant bone and soft tissue tumor, in the 1890s [13]. He injected streptococcal organisms into his patient with cancer to make the patient infected and stimulate their immune system. This therapy is known as Coley toxin, and this development was the first milestone of immunotherapy. Though the concept of cancer immunosurveillance was furthered by the efforts of Burnet and Thomas in the 1950s [14], these efforts and other approaches attempting to overwhelm cancer via immunological approaches failed in the following half century. Following this, Schreiber et al. developed the concept of cancer immunoediting, wherein the relationship between cancer and the immune system is separated into three distinct phases (Figure 1) [15]. The first phase is Elimination, which is the phase where the generated cancer is eliminated by immune cells. The second phase is Equilibrium, where the cancer—with low immunogenicity, having been edited by the immune system in the first phase—and immune cells attack each other in the Equilibrium state. Finally, in the Escape phase, the more edited cancer cells can avoid immune system elimination and proliferate [16]. In this theory, all cancers with clinical appearance are in the Escape phase, which means they have the ability to escape from immune attack. Accordingly, a more powerful method of attacking the cancer, such as high specificity, prominent killer ability, or invalidating the escape method, is needed.

### 2.1. Adaptive Immunity

In vertebrates, the immune system is separated into two main systems, the innate immune system and the adaptive immune system (Table 1). The adaptive immune system is a newer evolutionary defense strategy than the innate one and is characterized by its slow but highly specific and long-lasting reactions towards foreign matter [17]. The main players of this system are T-cells, which are divided into several subclasses [18]. In this chapter, we present the information about CAR T-cells, the Cancer vaccine, and the Dendritic cell (DC) vaccine.

#### 2.1.1. Cancer Vaccine

The development of the cancer vaccine was started in the 1970s using irradiated autologous tumor cells [19]. The expectation that a whole tumor cell potentially contains all antigens of the cancer is an important aspect of this strategy, and this type of vaccine is tested in various cancers [20]. Though this was not successful for human osteosarcoma, the autologous cancer vaccine platform for canine osteosarcoma has been developed and focused on recently [21]. Still, this approach has weaknesses, such as a threshold tumor volume needed for use as an antigen, resulting in limited applicability. To address this issue, an allogenic tumor vaccine was developed and tested [22,23,24,25], but no promising result was reported for osteosarcoma.

At the same time as the detection of tumor associated antigens (TAAs), protein/peptide-based vaccines were developed. The first trial using this method was performed after the detection of the first human cancer antigen, MAGE-1, by Boon et al. in 1991 [26]. To induce a cancer specific immune response, lots of TAAs are administered using varied forms such as cancer antigen proteins, epitope peptides, mRNA, and cDNA. For osteosarcoma, Tsukahara et al. reported papillomavirus binding factor (PBF) as human autologous osteosarcoma gene [27] and Tsuda et al. reported SART3 as a promising candidate for immunotherapy for osteosarcoma patients [28]. On the other hand, Wilms’ tumor gene 1 (WT1), which is reported to have relationship with osteosarcoma [29,30], is also focused on as a candidate for peptide use in vaccines [31]. Research seeking a new antigen using methods such as proteomics is ongoing [32,33].

The method using sensitized antigen presenting cells (APCs) such as DCs is also categorized as a cancer vaccine.

#### 2.1.2. DC Vaccine

DCs are a type of APCs [34], and stimulated DCs from a certain antigen are used as a cancer vaccine (Figure 2). In the first generation of DC vaccine, the hematopoietic stem cells and monocytes, which are progenitor cells of DCs, are collected from the patient, differentiated to DCs by stimulating with granulocyte-macrophage colony-stimulating factor (GM-CSF), cocultured with antigen, and the stimulated mature DCs are then returned to the patient [35]. In this method, the safety and induction of tumor-specific T-cell responses are proven [36,37], and promising phase III studies on brain cancer, colorectal cancer, and melanoma are now ongoing (NCT00045968, NCT02503150, NCT01983748, respectively). For osteosarcoma, Camille et al. reported that the DC vaccine delayed osteosarcoma progression or induced tumor regression in a rat osteosarcoma model [38]. In the clinical trial, however, only a limited effect was observed in two cohorts using tumor cell lysate as an antigen for recurrent osteosarcoma [39,40]. DC vaccine clinical trials with pretreatment of Decitabine (NCT01241162) or Gemcitabine (NCT01803152) are currently being conducted.

On the other hand, to stimulate a large number of DCs, the in vivo DC vaccine is preferred to the ex vivo DC vaccine. The key finding that makes the in vivo DC vaccine possible is the DC function of the cross presentation. Major histocompatibility complex (MHC) class I presents cytosolic peptides and can stimulate CD8+ cells if the peptides derive from the non-self, such as a viral product. On the other hand, MHC class II presents peptides derived from extracellular antigen protein taken up via phagocytosis and can stimulate CD4+ cells. DCs can present extracellular peptides not only in MHC class II but also in MHC class I using cross presentation, which means it is possible to directly stimulate CD8+ and CD4+ T-cells using an extracellular antigen. Using these characteristics, an antigen combined with an anti-DC specific antigen antibody can be effectively delivered to DCs and activate both CD4+ and CD8+ T-cells after DC maturation with an adjuvant such as TLR3 or CD40 agonist [41]. 

#### 2.1.3. CAR T-cells

In the 1980s, Rosenberg et al. used interleukin (IL)-2 as a stimulus for lymphoid cells and showed its ability to lyse cancer cells in vitro [42]. This method is called Lymphokine Activated Killer (LAK) cell therapy and was the first trial of the adoptive immunotherapy. In the concept of adoptive immunotherapy, immune cells taken from patient are expanded with/without genetic modification ex vivo and returned to the patient with the intent to kill the cancer specifically. To increase specificity and efficacy, CAR T-cells are used for the adoptive immunotherapy [11]. CAR T-cells are T-cells expressing CARs genetically containing the recognition domain by using a component of an antibody for TAAs called a single chain variable fragment (scFv) and can be activated in a tumor specific manner. After generating the first generation CARs [43], which simply consist of scFvs, the trans membrane domain, and intracellular T-cell signaling domain (CD3ζ), second and third generation CARs, which contain one or more co-stimulatory domains (CD), and forth generation CARs with a component of an inducible transgenic product including pro-inflammatory cytokines (IL-12) to activate CAR T-cells, were developed [11]. T-cells with two or more different CAR expressions are called multi-targeting CAR T-cells [44], and the next generation of CARs is being developed (Figure 3) [45].

In designing each generation of CAR T-cell, the choice of target antigen is crucial. For example, the most successful and well-known CAR T-cell therapy targeted CD19 in acute lymphoblastic leukemia (ALL) [46]. For osteosarcoma, there are several promising target antigens which have been used for clinical trials. Human epidermal growth factor receptor (HER) 2 is well known as an oncogenic factor and is routinely tested in patients with breast cancer [47]. Anti-HER2 antibodies such as trastuzumab have been developed and have significantly improved the outcomes of HER2 positive breast cancer patients [48]. Though trastuzumab is not as effective as osteosarcoma treatment [49], HER2 CAR T-cells are still promising, because HER2 CAR T-cells have an ability to recognize and kill cells with lower HER2 expression [50]. Indeed, several in vitro and in vivo studies using xenograft models show the therapeutic ability of HER2 CAR T-cells for osteosarcoma. For clinical use, Ahmed et al. reported a phase I/II study using second generation CAR T-cells for patients with HER2 positive recurrent/refractory sarcoma in a cohort containing 16 patients with osteosarcoma [51]. In this study, except for one patient that developed fever within 12 hours after CAR T-cell infusion treated by ibuprofen, no major complications were observed. On the other hand, in terms of the outcome, except for two non-evaluable patients and three patients with stable disease for 12–15 weeks, 11 patients were evaluated as having progressive disease. The newer generation of CARs or any lymphodepleting regimen may help to improve the outcome. As well as the HER2 CAR T-cells, Disialoganglioside (GD2) [52], Interleukin 11 receptor alpha subunit (IL-11RA) [53], Insulin-like growth factor 1 receptor (IGF-1R) with receptor tyrosine kinase orphan-like receptor 1 (ROR1) [54], and B7-H3 [55] are reported as good targets in vitro or in vivo using mouse studies. Several GD2 CAR T-cell clinical trials (NCT02107963, NCT01953900, NCT03635632, NCT03356782) were conducted, and the results are yet to be published.

### 2.2. Innate Immunity

In contrast to the adaptive immune system, the innate immune system is an older system, and the most important characteristic of this system is to show immune response for cancer via TNF-α and Fas–Fas ligands without stimulation by the antigen [17]. The main players of this system are Natural killer (NK) cells, but in terms of T-cell related cancer immunotherapy, the γδ T-cells contribute a unique and important role.

The γδ T-cells are defined as T-cells with γδ T-cell receptors (TCRs) [56] instead of αβ TCRs with “normal” T-cells and consist of approximately 4% of T-cells [57]. Though many aspects of γδ T-cells are not fully understood, the characteristics of γδ T-cells are thought to be low antigen specificity, non-MHC restriction, high cytokine secretion, and quick response [58]. In other words, γδ T-cells are thought to act like members of the innate immune system [57]. In terms of cancer immunity, some tumor cells hide the expression of MHC to escape attack from T-cells because T-cells usually need MHC to activate themselves, but γδ T-cells do not need the signal from MHC and can attack the tumor cells without MHC expression [59]. Vγ9/Vδ2 T-cells, which form the majority of γδ T-cells in human beings, recognize isopentenyl pyrophosphate (IPP) in the mevalonate pathway and activate γδ T-cells showing strong cytotoxic ability towards cancers [60]. It is also well known that zoledronate with IL-2 can activate γδ T-cells via accumulating IPP by inhibiting the mevalonate pathway [61]. 

In terms of cancer immunotherapy, Dieli et al. conducted γδ T-cell therapy by administrating zoledronate with IL-2 for hormone-refractory prostate cancer and reported good clinical course and tolerability [62]. Adoptive immune therapy using γδ T-cells was also conducted. In this strategy, collected γδ T-cells are stimulated, expanded, and maintained by zoledronate with IL-2 before being returned to patients [63]. Some phase I or II trials were conducted and reported promising results [64,65,66]. In terms of osteosarcoma, Kato et al. reported the lytic ability of γδ T-cells in osteosarcoma cell lines stimulated by pamidronate and IL-2 [60]. Muraro et al. reported the suppressive ability of γδ T-cells in osteosarcoma cell lines using zoledronate as a stimulus [67]. Recent studies focused on combination strategies with zoledronate and decitabine [68] or valproic acid [69].

### 2.3. Immune Checkpoint Inhibitor

To maintain the appropriate level of immune response or prevent over-inflammatory conditions, the immune system has a natural immune suppressive function. One of the functions is the immune checkpoint, and many molecules are immune checkpoint molecules, such as Cytotoxic T-Lymphocyte Associated Protein 4 (CTLA-4), Programmed cell death 1 (PD-1), T-cell immunoglobulin, and mucin domain-containing protein 3 (Tim-3). T-cells express these molecules and suppress their activity when they bind with ligands, even if the activation signal via MHC is already activated. In normal tissue, the function is seen in immune suppressive cells such as regulatory T-cells (Treg) or Myeloid-derived suppressor cells (MDSCs) and effector T-cells or APCs. In the cancer microenvironment, cancer cells secrete chemokine (e.g., CCL20) to recruit Treg and also express the immune checkpoint molecule to suppress the immune response. By using the ICI, “exhausted” T-cells are released from these suppressions and recover the ability to attack the cancer (Figure 4) [70]. The first report of the anti-cancer ability of the ICI was written in 1996 by Leach et al. [71]. In this study, in vivo administration of antibodies to CTLA-4 resulted in the rejection of colon carcinoma and fibrosarcoma transplanted to mice. By using an anti-CTLA4 antibody, the Treg expressing CTLA4 for their suppressive function were eliminated via antibody-dependent cellular cytotoxicity (ADCC) [72], and suppressed T-cells could be re-activated [73].

In a 2010 clinical trial, Hodi et al. reported that the anti-CTLA-4 antibody, Ipilimumab, improved survival in patients with metastatic melanoma [74]. In 2012, another immune checkpoint inhibitor anti-PD-1 antibody showed objective responses in patients with non-small-cell lung cancer, melanoma, or renal-cell cancer [75]. These reports were considered as a breakthrough in immune therapy for cancers.

In the context of osteosarcoma, immune checkpoint inhibitors are thought to be promising [76] because the number of tumor infiltrated lymphocytes (TILs) in osteosarcoma is known to be much larger than for other sarcomas [12], meaning that many more TILs can be utilized by using immune checkpoint inhibitors.

The expression of the ligand of PD-1 (i.e., PD-L1) on the surface of osteosarcoma was reported [77], and in our research, interferon gamma (IFN-γ), which is one of the inflammatory cytokines, increases the expression of PD-L1 on the surface of osteosarcoma in vitro [78]. In this study, in which the anti-PD-1 antibody was used for an osteosarcoma transplanted mouse model, the suppressive effect on tumor growth and the improvement on overall survival were shown.

In the clinical setting, PD-L1 expression is known to correlate with bad prognosis [79], and patients with higher levels of T-cell activation markers show good prognosis [80]. However, the phase 2 clinical trial using the anti-PD-1 antibody (Pembrolizumab) with sarcoma patients resulted in only one of the 22 patients with osteosarcoma having an objective response [81]. In this trial, the Response Evaluation Criteria in Solid Tumors version 1.1 (RECIST 1.1) [82] was used as an indication tool to evaluate the efficacy and the definition of “objective response” is patients with complete or partial response. The opinion that RECIST 1.1 may not be suited to evaluating immunotherapy nor study for osteosarcoma should be considered. In other words, though the change of tumor size is mainly used to judge the efficacy in RECIST 1.1, the tumor size increases in immunotherapy even if the therapy is effective because of the infiltration of immune cells in the tumor site [83]. Though the validity of this opinion is controversial [84,85,86], specialized evaluation tools for immunotherapy such as irRC [87], irRECIST [88], and iRECIST [83] were developed to address the problem. On the other hand, osteosarcoma consists of a large amount of extracellular matrix, and it is difficult to shrink the tumor in terms of its size even if the therapy is effective. Because of this, the necrosis rate after chemotherapy is traditionally used to evaluate the efficacy of chemotherapy for osteosarcoma [89], and pathological evaluation may be needed for immunotherapy as well. There are some clinical trials currently being conducted using immune checkpoint inhibitors with osteosarcoma patients (NCT03006848, NCT04044378, NCT02982486, NCT03013127, NCT04294511, NCT03676985, NCT04359550, NCT03628209, NCT03277924, NCT03359018, NCT04351308, NCT02500797, NCT02815995, Table 2). One of these studies is a phase II trial of Avelumab (NCT03006848) that adopts cell proliferation as one of the outcome measures, and results are yet to be published.

## 3. Challenges for the Future

We briefly reviewed T-cell related immunotherapy for osteosarcoma. Though some promising strategies have been developed, there is no immunotherapy firmly proven to affect osteosarcoma yet, especially in clinical use. Each form of immunotherapy has its strong points and weak points. For example, unlike vaccines for infection, the cancer vaccine administers an antigen that potentially pre-exists in the patient. According to the concept of immunoediting, the appearing cancer gains escape methods from immune attack, which means that monotherapy using a cancer vaccine that simply loads much of the “pre-existing” antigen may not be sufficient for cancer therapy. Thus, for example, treatments targeting immune tolerance may need to be used together. This issue is just as valid for CAT T-cell therapy, and in this context, combination therapy with the ICI seems promising.

On the other hand, the cause of the insufficient effect of ICI monotherapy for osteosarcoma has been gradually revealed. Wu et al. conducted whole genome, RNA, and T-cell receptor sequencing; immunohistochemistry; and reverse phase protein array profiling on 48 osteosarcoma specimens. According to this report, the median immune infiltration level was lower than in other tumor types where the ICI is effective with concomitant low T-cell receptor clonalities. The original antigen of each cancer case’s (neoantigen) expression in osteosarcoma was lacking and significantly associated with high levels of nonsense-mediated decay. Samples with low immune infiltration had a higher number of deleted genes including MHC, while those with high immune infiltration expressed higher levels of adaptive resistance pathways. They concluded that these multi-resistant pathways inhibit the effect of immunotherapy for osteosarcoma (Figure 5 [90]).

From another perspective, the promising approaches mentioned in this article seem suitable for these characteristics of osteosarcoma as good strategies. Thus, combination therapy of the ICI with another immunotherapy seems promising. Indeed, there are several types of immunotherapy combination therapies. Ladle et al. conducted combination therapy of the anti-PD-1 antibody and cancer vaccine for a mouse model of osteosarcoma and reported complete rejection of tumor in 70% of mice [91]. In the clinical trial, a DC vaccine (MASCT-I) with anti-PD-1 antibody (NCT04074564) is planned.

Another important issue is the correct evaluation of the mechanisms of immunotherapy. This is required in order to understand the essential mechanisms of certain immunotherapies because the immune system is complex and not fully understood. For example, we reported the phenomenon of tumor infiltrating Treg decreasing after administering anti-PD-1 antibody in an osteosarcoma cell line transplanted mouse model and considered it to be the result of ADCC with Treg expressing PD-1, as the same as the anti-CTLA4 antibody [78]. If this conclusion is true, the anti-PD-1 antibody possesses the ability to not only inhibit the PD-1/PD-L1 axis [92] but also to deplete Treg, and it could explain why the PD-L1 expression on the surface of the tumor does not directly correlate with the response of the anti-PD-1 antibody therapy or why the efficacy of the anti-PD-1 antibody is superior to that of the anti-PD-L1 antibody [93] (though the efficacy of the anti-PD-1 and anti-PD-L1 antibodies is also controversial [94,95]). The discussion of the relationship between the anti-PD-1 antibody and Treg is still controversial, and a report with the opposite result with a squamous cell carcinoma model [96] also exists. One more example used zoledronate, tested for use in osteosarcoma as an antitumor drug, but the efficacy was very low [97], though use with IL-2 reportedly stimulated γδ T-cells in an orthotopic mouse model of osteosarcoma. Thus, it is important to evaluate the mechanism of therapy correctly. In other words, more studies are needed on immunotherapy, and an examination of the results and phenomena from a diversified viewpoint may be important.

## 4. Conclusions

Immunotherapy has a long history, but, especially for osteosarcoma, it has not been necessarily effective. Recently, some promising approaches using T-cells have been developing, and further investigations including clinical trials are now ongoing. We believe the accumulation of knowledge helps us to offer the best treatment, including immunotherapy, for osteosarcoma patients.

## Figures and Tables

**Figure 1 ijms-21-04877-f001:**
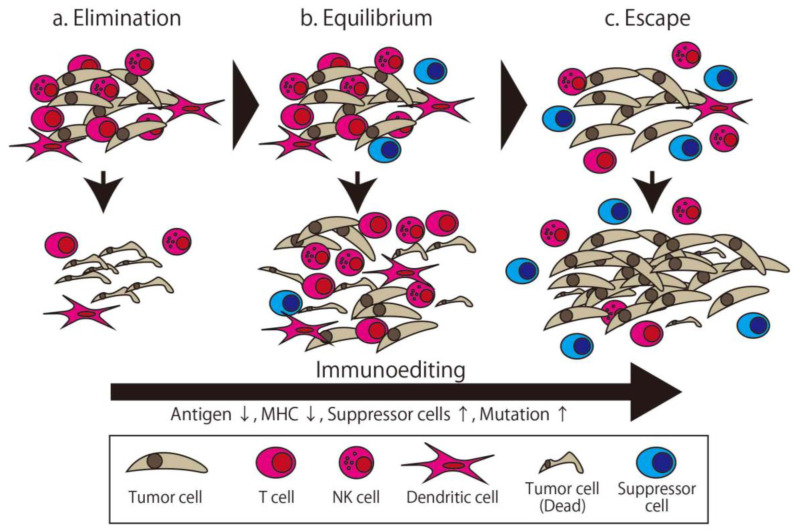
The three phases of cancer immunoediting. The tumor is gradually edited to gain resistance to immune attack. (**a**) In the Elimination phase, the tumor is eliminated by the immune attack. (**b**) In the Equilibrium phase, some of the edited tumor cells survive and are eliminated incompletely. (**c**) In the Escape phase, highly edited tumor cells can proliferate. The apparent clinical cancer is in the Escape phase.

**Figure 2 ijms-21-04877-f002:**
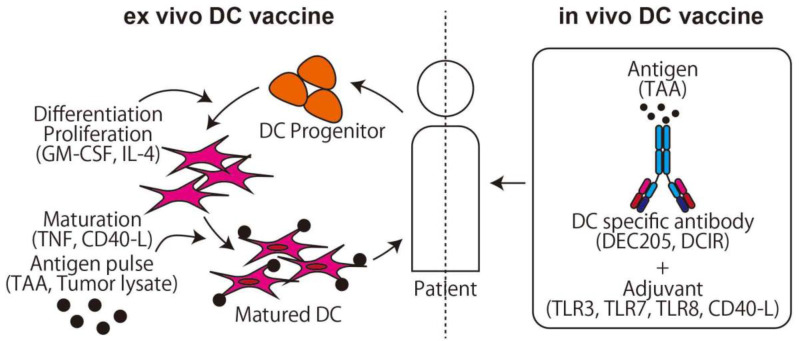
The two main methods of dendritic cell (DC) vaccine. The ex vivo DC vaccine uses the method of adoptive immunotherapy. The in vivo DC vaccine uses a specific antibody conjunction with an antigen that targets DCs and activates T-cells via cross presentation.

**Figure 3 ijms-21-04877-f003:**
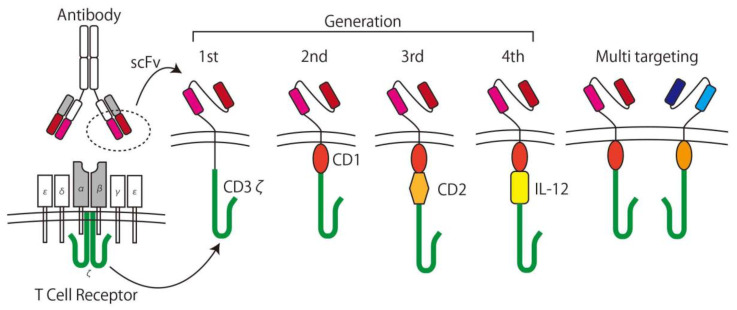
The development of chimeric antigen receptor (CAR) T-cells. A CAR consists of a single chain variable fragment (scFv) as the antigen recognition domain and CD3ζ as the T-cell signaling domain. Later generation CARs have at least one co-stimulating domain (e.g., CD27, CD28, CD134, CD137) and cytokine inducible domains (e.g., IL-12). Multi-targeting CARs have two or more sets of CARs on the surface of the T-cell.

**Figure 4 ijms-21-04877-f004:**
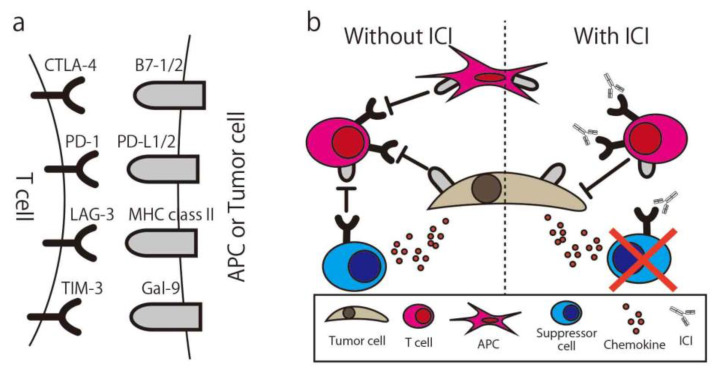
Immune Checkpoint Molecule and Immune Checkpoint Inhibitor. (**a**) The combination of common immune checkpoint molecules and their ligands. (**b**) Simplified illustration of the immune checkpoint inhibitor mechanism. CTLA-4, Cytotoxic T-Lymphocyte Associated Protein 4; PD-1, Programed cell death 1; LAG-3, Lymphocyte Activating 3; TIM-3, T-cell immunoglobulin mucin-3; Gal-9, galectin-9; ICI, Immune checkpoint inhibitor; APC, antigen-presenting cell.

**Figure 5 ijms-21-04877-f005:**
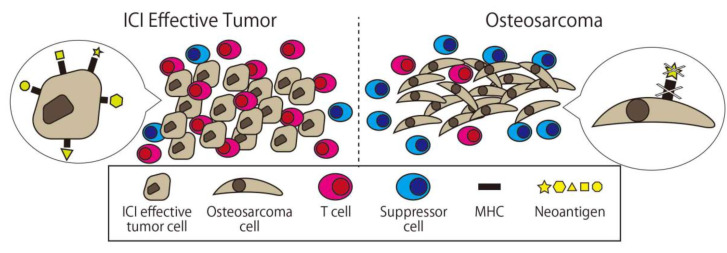
Comparison between osteosarcoma and ICI effective tumor cells. There are fewer immune cells and more suppressors cell in osteosarcoma, and there is less expression of major histocompatibility complex (MHC) and neoantigen on the surface of osteosarcoma.

**Table 1 ijms-21-04877-t001:** The difference between the innate and the adaptive immune system.

	Innate	Adaptive
Specificity	Non-specific	Specific
Response	Rapid	Slow
Memory	No	Yes
Main Players	NK cell Macrophage Granulocyte	T-cell B cell

**Table 2 ijms-21-04877-t002:** Registered Clinical Trials using T-cell Related Immunotherapy for Osteosarcoma.

Trial ID	Type of Immunotherapy	Target Disease	Techniques	Phase	Status
NCT01241162	DC vaccine	Osteosarcoma, other cancer	DC vaccine, Decitabine	1	Completed without result
NCT01803152	DC vaccine	Osteosarcoma, other sarcoma	DC vaccine, Gemcitabine	1	Active, not recruiting
NCT02107963	CAR T-cell	Osteosarcoma, other cancer	GD2-CAR, AP1903, Cyclophosphamide	1	Completed without result
NCT01953900	CAR T-cell	Osteosarcoma, Neuroblastoma	GD2-CAR, VZV vaccine, Fludarabine, Cyclophosphamide	1	Active, not recruiting
NCT03635632	CAR T-cell	Osteosarcoma, other cancer	C7R-GD2-CAR, Fludarabine, Cyclophosphamide	1	Recruiting
NCT03356782	CAR T-cell	Osteosarcoma, other sarcoma	Each sarcoma specific CAR-T-cell	1/2	Recruiting
NCT03628209	Anti-PD-1 antibody	Osteosarcoma	Nivolumab, Azacitidine, surgery	1/2	Recruiting
NCT03277924	Anti-PD-1 antibody	Osteosarcoma, other sarcoma	Nivolumab, Sunitinib	1/2	Recruiting
NCT04294511	Anti-PD-1 antibody	Osteosarcoma	Camrelizumab, Neoadjuvant chemotherapy	2	Recruiting
NCT03359018	Anti-PD-1 antibody	Osteosarcoma	SHR-1210, Apatinib	2	Completed
NCT04351308	Anti-PD-1 antibody	Osteosarcoma	Camrelizumab, MAPI, Apatinib	2	Recruiting
NCT03013127	Anti-PD-1 antibody	Osteosarcoma	Pembrolizumab	2	Active, not recruiting
NCT04044378	Anti-PD-1 antibody	Osteosarcoma	Camrelizumab, Famitinib, Isosfamide	2	Withdrawn (Toxicity)
NCT03676985	Anti-PD-L1 antibody	Osteosarcoma	ZKAB001	1/2	Recruiting
NCT04359550	Anti-PD-L1 antibody	Osteosarcoma	ZKAB001	3	Not yet recruiting
NCT03006848	Anti-PD-L1 antibody	Osteosarcoma	Avelumab	2	Active, not recruiting
NCT02500797	Anti-PD-1 a/o L1 antibody	Osteosarcoma, other cancer	Nivolumab, Ipilimumab	2	Active, not recruiting
NCT02982486	Anti-PD-1 a/o L1 antibody	Osteosarcoma, other cancer	Nivolumab, Ipilimumab	2	Not yet recruiting
NCT02815995	Anti-PD-1/L1 antibody	Osteosarcoma, other sarcoma	Durvalumab, Tremelimumab	2	Active, not recruiting
NCT04074564	DC vaccine, Anti-PD-1 antibody	Osteosarcoma, other sarcoma	MASCT-I, anti-PD-1 antibody, Apatinib	1	Not yet recruiting

VZV, varicella-zoster virus; a/o, and/or.

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
