# Peer review of "A Review of T-Cell Related Therapy for Osteosarcoma"

_ijms, 2020, doi:10.3390/ijms21144877_

Round 1

Reviewer 1 Report

Chapter 1. (Introduction):  The authors described history and current treatment of osteosarcoma. Special attention on T-cell-related therapies was also provided. In the next chapters (2. Cancer immune therapy and cancer immunoediting, 3.1 Adaptive immunity, 3.2. Innate immunity, 3.3. Immune checkpoint inhibitor) the authors widely described immunotherapy and strategies for many cancers over the last ten years. However, in some places (chapters: 2, 3.1, 3.2, 3.3) descriptions of methods of treating cancers other than osteosarcoma are too extensive and overwhelm the main topic of the article. Chapter 4 (Challenges for the future) summarizes current treatment methods and gives a perspective on future treatment methods, which are concluded in chapter 5.

General comments: “A review of T-cell related therapy for osteosarcoma” - is an interesting, clear, concise, and well-written review article. 

The layout of the article: correct.

References selection: wide.

English level: appropriate and understandable.

Specific comments: none.

Overall recommendation: accept in present form.

Author Response

Thank you for your valuable comments for our paper.

I tried to minimize the excess information before submitting the first version, but I agree with your comment that some parts are still overwhelming. I guess you recommended ”accept in present form”’ because you noticed the difficulty to be balanced between simplify and accuracy. I follow your recommendation and thank you again for your deep understanding.

Reviewer 2 Report

The Authors collected the current knowledge regarding T-cell related immunotherapy in osteosarcoma. This is a timely and significant review on current and potential T-cell related immunotherapy for osteosarcoma.  This topic has not received a great deal of attention, despite its clear importance, what is also emphasized in this review.

The Authors made an excellent work - the review is comprehensive and provides sufficient knowledge also covered by clear figures.

Overall the review is interesting and well prepared.

Please reconsider stylistics in some sentences, several examples:

Though Burnet and Thomas advocated the concept of cancer immunosurveillance, in which said immune system destroys the cancer tissue and  suppresses its development, in the 1950s [14], and many efforts were made to overwhelm cancer via  immunological  approaches,  most  of  them  failed  in  the  following  half  century.

To resolve the problem, an allogenic tumor vaccine was developed  and tested for some cancers [22-25], but, for osteosarcoma, no promising result was reported.  

The stages of cancer immunoediting are sometimes written with small letters and sometimes with upper. Same with cancer vaccines – please unify.

Before publishing Authors should consider the edition of the manuscript, as many of the sentences and even the structure of the paragraphs are not consistent and hard to follow.

Author Response

Thank you very much for your positive comments on our manuscript.

According to your advice, I made some changes in the manuscript.

I wrote the detail below.

I changed some sentence according to great advice from English native speaker because I have used MDPI English editing service before the first submission and I thought I need the help to satisfy your recommendation.

So, I would like to add the person’s name in the acknowledgement section.

[Modifications have been made to page 10 (line 320)]

I added the sentence “We are grateful to Ramsey Bekdash for English writing assistance.”

Please reconsider stylistics in some sentences

I changed some sentence or words to improve stylistics and make it easy to understand.

I write the changes down below.

[Modifications have been made to page 1 (line 32)]

I change the sentence to “One reason for this”.

[Modifications have been made to page 1 (line 41-44)]

I changed the sentence to “For osteosarcoma especially, these therapeutic options are promising as it has been reported that the number of tumor infiltrating T-cells is greater than that of other types of sarcoma [12]. Because of this, many immune therapies are being trialed in pre- and post-clinical settings.  ”

[Modifications have been made to page 2 (line 54-59)]

I changed the sentence to “Though the concept of cancer immunosurveillance was furthered by the efforts of Burnet and Thomas in the 1950s  [14], these efforts and other approaches attempting to overwhelm cancer via immunological approaches failed in the following half century. Following this, Schreiber et al. developed the concept of cancer immunoediting, wherein the relationship between cancer and the immune system is separated into three distinct phases (Fig. 1)”

[Modifications have been made to page 2 (line 59-63)]

I changed the sentence to “The first phase is Elimination, which is the phase where the generated cancer is eliminated by immune cells. The second phase is Equilibrium where the cancer with low immunogenicity, having been edited by the immune system in the first phase, and immune cells attack each other in the Equilibrium state. Finally in the Escape phase, the more edited cancer cells can avoid immune system elimination and proliferate”

[Modifications have been made to page 3 (line 85)]

I changed the sentence to “Though this was not successful”

[Modifications have been made to page 3 (line 86-89)]

I changed the sentence to “Still this approach has weaknesses, such as a threshold tumor volume needed for use as an antigen, resulting in limited applicability. To address this issue, an allogenic tumor vaccine was developed and tested [22-25], but no promising result was reported for osteosarcoma.”

[Modifications have been made to page 5 (line 152-155)]

I changed the sentence to “For osteosarcoma, there are several promising target antigens which have been used for clinical trials.”

[Modifications have been made to page 10 (line 309-310)]

I changed the sentence to “it has not been necessarily effective”

The stages of cancer immunoediting are sometimes written with small letters and sometimes with upper.

[Modifications have been made to page 2 (line 69-72)]

I unified to write the stage name with upper case.